# Malnutrition and Increased Risk of Adverse Outcomes in Elderly Patients Undergoing Elective Colorectal Cancer Surgery: A Case-Control Study Nested in a Cohort

**DOI:** 10.3390/nu14010207

**Published:** 2022-01-03

**Authors:** Cristina Martínez-Escribano, Francisco Arteaga Moreno, Marcos Pérez-López, Cristina Cunha-Pérez, Ángel Belenguer-Varea, David Cuesta Peredo, Francisco Javier Blanco González, Francisco J. Tarazona-Santabalbina

**Affiliations:** 1Anesthesiology and Resuscitation, Hospital Universitario de la Ribera, 46600 Valencia, Spain; martinez_criesc@gva.es; 2School of Doctorate, Catholic University of Valencia, San Vicente Martir, 46600 Valencia, Spain; francisco.arteaga@ucv.es (F.A.M.); marcos.perez@mail.ucv.es (M.P.-L.); cristina.cunha@ucv.es (C.C.-P.); belenguer_angvar@gva.es (Á.B.-V.); cuesta_davper@gva.es (D.C.P.); 3Division of Geriatrics, Hospital Universitario de la Ribera, 46600 Valencia, Spain; 4Quality Management, Hospital Universitario de la Ribera, 46600 Valencia, Spain; 5General and Digestive Surgery, Hospital Universitario de la Ribera, 46600 Valencia, Spain; blanco_fragon@gva.es; 6Centro de Investigación Biomédica en Red Fragilidad y Envejecimiento Saludable (CIBERFES), 28029 Madrid, Spain

**Keywords:** colorectal surgery, malnutrition, ERAS, postoperative complications, older patients

## Abstract

Background: Malnutrition increases worse outcomes during hospital admission for elective colorectal cancer (CRC) surgery in older adults. Methods: This work was designed an observational, monocentric, case-control study nested in a cohort of patients undergoing elective surgery for CRC disease at the Hospital Universitario de la Ribera (HULR) (Alzira, Valencia, Spain) between 2011 and 2019. The study considered patients with a CONUT score in the range of moderate to severe malnutrition (>4 points), with control patients with normal nutritional situations or mild malnutrition. Results: Moderate-to-severe malnutrition cases presented a greater length of stay (LOS), a higher incidence of adverse events (both medical and surgical complications), a higher incidence of surgical-wound infection, a greater need for blood transfusion, and a greater amount of transfused packed red blood cells. During hospitalization, the percentage of patients without nutritional risk decreased from 46 to 9%, and an increase in mild, moderate, and severe risk was observed. Patients with severe nutritional risk at hospital admission had significantly increased mortality at 365 days after discharge (HR: 2.96 (95% CI 1.14–7.70, *p* = 0.002)). After adjusting for sex, age, and Charlson index score, patients with severe nutritional risk at admission maintained a higher mortality risk (HR: 3.08 (95% CI 1.10–8.63, *p* = 0.032)). Conclusion: Malnutrition prevalence is high in older adults undergoing CRC elective surgery. Furthermore, this prevalence increases during hospital admission. Malnutrition is linked to worse outcomes, such as LOS, surgical and clinical complications, and mortality. For this reason, nutritional interventions are very important in the perioperative period

## 1. Introduction

Colorectal cancer (CRC) is the third most diagnosed cancer and the second leading cause of death from cancer worldwide [1]. In fact, CRC is the second most prevalent type of cancer in the world, with over 1.4 million cases and 693,900 deaths a year [2]. Age, genetics, and environmental factors, such as obesity, sedentary lifestyle, red meat and processed meat, tobacco and alcohol consumption, diabetes mellitus, and insulin resistance are associated with the development of CRC [3]. Surgery plays an important role, as it is the most effective treatment to cure this condition. [4] However, surgery is associated with a high rate of complications, ranging from 8% to 63% [5], and an overall perioperative mortality of between 1% and 12% [6]. 

As previously mentioned, advanced age is associated with greater perioperative mortality, a higher rate of perioperative complications, and higher costs [7]. Malnutrition is a powerful predictor of morbidity, mortality, long-term hospitalization, and readmission [8]. Malnutrition can be defined as an unbalanced nutritional state that compromises body reserve and function [9]. Rates of malnutrition in colorectal cancer patients range from 20% to 37% [10], depending on the tool used to assess nutritional status. Additionally, malnutrition affects treatment tolerability and postoperative complications, including anastomotic leakage (AL) and oncological outcomes [11].

Controlling nutritional status (CONUT) score [12] is an index that allows for assessment of nutritional condition. CONUT score is calculated from serum albumin, total cholesterol concentration, and peripheral lymphocyte counts. CONUT score is a prognostic factor of postoperative complications [13] and mortality in patients with CRC that useful for estimation of preoperative risk [14]. However, few studies have focused on complications occurring during the perioperative period in patients undergoing elective surgery for CRC. For this reason, our aim was to study the link between nutritional condition assessed with the CONUT tool and perioperative outcomes in patients undergoing elective surgery for CRC disease.

## 2. Materials and Methods

### 2.1. Study Design and Subjects

An observational, monocentric, case-control study nested in a cohort of patients undergoing elective surgery for CRC disease at the Hospital Universitario de la Ribera (HULR) (Alzira, Valencia, Spain) between 2011 and 2019 was designed. HULR is a tertiary-care hospital providing healthcare to a population of 253,330 inhabitants, of which 13.5% are over 69 years of age. 

A patient with a CONUT score in the range of moderate to severe malnutrition (more than 4 points) was considered a case patient, and a patient with a normal nutritional status or mild malnutrition (CONUT equal to or less than 4 points) was considered a control patient.

### 2.2. Eligibility Criteria

Inclusion criteria: patients aged 70 years or older undergoing elective open or laparoscopic colorectal surgery in cancer stage I-III at diagnosis time.

Exclusion criteria: emergency surgery; presence of metastases; patients operated on in other centers and referred to the HULR for sectorization; palliative surgery; life expectancy less than 6 months according to the palliative prognosis score [15].

### 2.3. Sample Size

Data were collected from 371 patients enrolled consecutively, not selected, who underwent elective surgery for CRC between the above-mentioned dates. The calculated power of the study with the sample obtained, with an alpha error of 5% and a magnitude of effect between groups of 20%, was 97%.

### 2.4. Outcome Measures

#### Study Variables

The following demographic variables were collected: age and sex; anthropometric variables; frailty, measured with criteria established by Balducci [16]; other geriatric syndromes; comorbidities and Charlson index [17]; ASA physical classification status system [18]; tumor staging, blood-test results; and hospital process data, such as admission to the intensive care unit (ICU), length of stay, number of reinterventions, readmissions, and mortality.

Other variables included were the Fast-track or Enhanced Recovery After Surgery (ERAS^®^) protocol assessment in the preoperative period, consisting of anemia management and dietary advice to patients with nutritional deficiency [19]. A rehabilitation specialist conducted respiratory rehabilitation and provided a physical-activity regime tailored to each patient’s condition.

The primary outcome was to determine whether a poorer nutritional condition would increase the incidence of complications and adverse events, such as medical (delirium, infections, etc.) and surgical complications (suture dehiscence, paralytic ileus [defined as a lack of transit and oral tolerance established 5 days after surgery]) during hospital admission. Other objectives of the study were to assess the impact of nutritional status at admission and discharge on reinterventions, hospital stay, 30-day readmission rate, and in-hospital and 1-year mortality rate.

### 2.5. Statistic Analysis

The data obtained from the clinical history were analyzed with the statistical software program SPPS, version 23 (SPPS Inc. Chicago, IL, USA). Qualitative variables (including dichotomous variables) were described using absolute and relative frequencies. For quantitative variables, measures of central tendency (mean) were used, along with measures of dispersion (standard deviation, SD). A bivariate calculation was performed for the variables considered in the main and secondary objectives with Student’s t test for quantitative variables, with normal distribution, and with the Chi-square technique for qualitative variables. A multivariate analysis was performed using binary logistic regression for the main variable, calculating the crude and adjusted odds ratio (OR). A Cox regression was performed for analysis of mortality at 365 days based on the categorized score of the CONUT score at admission and at hospital discharge, calculating the crude and adjusted hazard ratio (HR). Finally, survival curves were calculated using Kaplan-Meier analysis for the same categories used in the Cox regression. The significance threshold was established at a value of *p <* 0.05.

### 2.6. Ethical Considerations

The study complied with legal requirements and good clinical practice guidelines, as well as the Declaration of Helsinki (updated October 2008 version of the World Medical Association on ethical principles for medical research in humans). The protocol was approved by the HULR Ethics and Clinical Research Committee (registry code HULR06112019).

## 3. Results

### Subject Characteristics

A total of 98 patients were considered cases, with 227 controls, according to CONUT score. Cases showed significantly higher comorbidity, as estimated by the Charlson index, with no differences found in age, sex, or anthropometric measures, such as body mass index, tumour stage, ASA score, or frailty. Likewise, the proportion of patients with previous FAST-TRACK or ERAS (enhanced recovery after surgery) evaluation was significantly higher in patients who presented a better nutritional situation on admission (Table 1).

Cases presented a longer hospital stay, a higher incidence of adverse events (both medical and surgical complications), a higher incidence of surgical-wound infections, a greater need for blood transfusion, and a greater amount of transfused packed red blood cells versus control group (Table 2). No statistically significant differences were observed in the incidence of delirium, suture dehiscence, readmission rate, or in-hospital and 365-day mortality. 

A binary logistic regression with variables showing statistical significance in the bivariate analysis was performed. Table 3 and shows crude odds ratios and Charlson index ERAS odds ratios (OR) by age and sex with CONUT score a hospital admission and discharge (Table 4), respectively. All variables with significant difference in bivariate analysis maintained significance in the regression analysis.

During hospitalization. the percentage of patients with nutritional risk increased in all groups—mild. moderate. and sever—overall. from 9 to 46%. Therefore. mild nutritional impairment estimated by CONUT increased from 25 to 38%. moderate nutritional impairment increased from 21 to 40%. and severe nutritional impairment increased from 8 to 13%. *p*< 0.001 (Figure 1).

Bivariate CONUT score calculation was repeated with hospital discharge blood-test results. Patients with nutritional impairment at hospital discharge presented a higher incidence of paralytic ileus and urinary tract infection and a higher rate of readmission in intensive care units (ICU) (Table 5). Similarly to the previous analysis. when repeating the analysis using a logistic regression. variables maintained statistical significance.

Patients with severe nutritional risk at hospital admission presented a statistically significant increase in mortality at 365 days (HR: 2.96 (95% CI 1.14–7.70. *p* = 0.002)). but this significance was not observed in patients with severe nutritional deficit at hospital discharge (HR: 5.24 (95% CI 0.64–42.6. *p* = 0.053)). After adjusting for sex. age. and Charlson index score. patients with severe nutritional risk at admission maintained statistical significance (HR: 3.08 (95% CI 1.10–8.63. *p* = 0.032)) with the most severe nutritional risk at hospital discharge reaching HR significance: 4.03 (95% CI 0.49–33.3. *p* = 0.195). Similarly. Table 6 and Table 7 and Figure 2A,B show the lower overall survival of patients with severe nutritional risk at hospital admission (*p* = 0.001) and at hospital discharge (*p* = 0.031).

## 4. Discussion

In our study. the prevalence of malnutrition estimated by CONUT score was high in older adults undergoing elective surgery for CRC. This prevalence increased significantly during hospital admission. Malnutrition during hospital admission was associated with a longer hospital stay and a higher incidence of adverse events. intensive care unit readmissions. and mortality.

CONUT is a simple and useful nutritional screening tool that has previously been used to estimate the nutritional status of patients undergoing elective surgery for CRC [12,13]. CONUT score has been associated with lower survival [13,20,21] and postoperative complications [12] in patients undergoing elective surgery for colorectal neoplasia.

However. few studies have focused on complications occurring during the perioperative period in patients undergoing elective surgery for CRC. In our study. we observed a significant increase in the prevalence of malnutrition from hospital admission to discharge. Preoperative CONUT score has previously been correlated with a higher mortality rate [22]. although not with increased length of stay. transfusion requirements. surgical-wound and urinary tract infections. or ICU readmissions observed in our study. Surgical-wound infection was associated with CONUT score at admission but not at hospital discharge. A recent study found that surgical-wound infection was linked to non-administration of preoperative enteral nutrition [23]. Nutritional intervention included in the ERAS program is crucial for the reduction of adverse events. such as surgical-wound infection.

Previous studies have reported that poorer nutritional condition is associated with a longer hospital stay and a higher incidence of wound-suture dehiscence [24]. as well as paralytic ileus [25]. However. CONUT has not been used as a nutritional screening tool. In our study. a higher incidence of paralytic ileus was observed with the CONUT score at hospital discharge but not at admission time. Regarding the study design. we were not able to report on whether the association between the CONUT score at hospital discharge and paralytic ileus is a cause or consequence. On the other hand. a statistically significant increased incidence of suture dehiscence was not observed. Patients with a CONUT score in the range of malnutrition had an increased risk of needing red-blood-cell transfusion. This relationship between poorer nutritional condition and a higher presence of anemia has been previously described [26] but not using CONUT as a nutritional screening tool.

Two variables influenced the nutritional status of older adults undergoing elective surgery for CRC: one negatively—comorbidity estimated by the Charlson index; the second. positively—prior assessment and intervention within the ERAS program. The Charlson index has been associated with poorer nutritional status [27]. while the ERAS program has been shown to improve the nutritional status of patients undergoing this surgery [28].

Despite the nutritional improvement described for the ERAS. we observed a significant deterioration in nutritional status during hospital admission. After less than 15 days of length of hospital stay. we found a very significant decrease in the percentage of patients without nutritional risk estimated by CONUT. Previous poor nutritional status has been previously described as a risk factor for bad outcomes during hospital admission for elective colorectal cancer surgery [8,29,30]. However. nutritional deterioration during hospital admission for elective colorectal cancer surgery had not been previously reported. These results suggest the need to intensify nutritional intervention during hospital admission.

The main limitations of the study are the long recruitment period (8 years). the greater comorbidity of the cases. estimated by means of the ICC. and the greater number of patients with ERAS intervention in the controls. For this reason. both variables have been used as adjustment variables in the logistic regression analysis. The percentage of lost data was calculated as 12%.

## 5. Conclusions

A poorer nutritional condition is correlated with a longer hospital stay. a higher incidence of complications. and a higher mortality. Nutritional condition worsens during the hospitalization of these patients and. compared to the nutritional situation on admission. is associated with a higher incidence of paralytic ileus and urinary tract infection. CONUT score is a useful nutritional screening tool in patients undergoing CRC elective surgery and can help to assess the nutritional evolution during hospital admission due to elective colorectal cancer surgery. ERAS could also reduce the prevalence of nutritional impairment in hospital admission. and these results suggest that intensification of this program before and during hospitalization could improve the nutritional status of these patients and reduce the incidence of complications and mortality rate.

## Figures and Tables

**Figure 1 nutrients-14-00207-f001:**
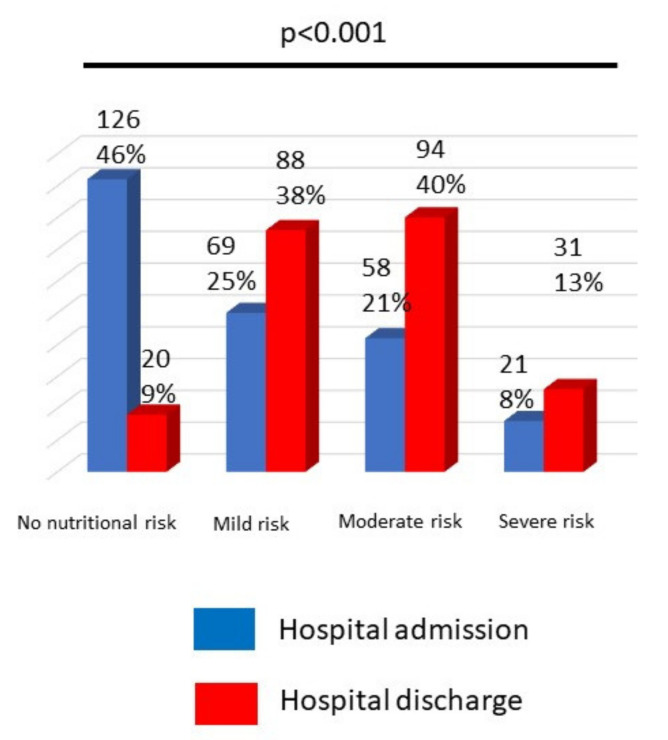
Changes in nutritional risk estimated by CONUT score.

**Figure 2 nutrients-14-00207-f002:**
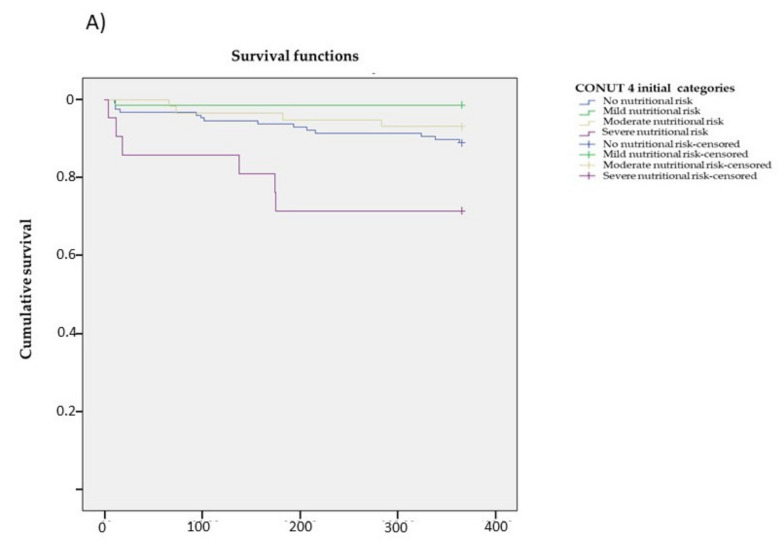
Survival curve using the Kaplan-Meier analysis based on the nutritional status estimated by CONUT. (**A**) Survival curve with the nutritional estimate using CONUT at hospital admission. **(B)** Analysis with the nutritional estimate using CONUT at hospital discharge.

**Table 1 nutrients-14-00207-t001:** Bivariate analysis of cases and controls during hospital admission. Baseline situation of cases and controls at hospital admission.

Variable.	Controls (*n* = 227)	Cases (*n* = 98)	*p*-Value
Age. mean (SD)	78.1 (5.4)	77.4 (5.2)	0.204
Gender, n(%)	♂ 123 (54%) ♀ 104 (46%)	♂ 62 (63%) ♀ 36 (37%)	0.144
BMI, Kg/m^2^	28.6 (4.5)	28.7 (5.1)	0.959
ASA	I 88 (39%) II 131 (58%) III 8 (3%)	I 37 (38%) II 58 (59%) III 3 (3%)	0.934
Tumour stage	I 64 (28%) II 72 (32%) III 91 (40%)	I 21 (21%) II 36 (37%) III 41 (42%)	0.261
Charlson Index	3.1 (3.2)	4.3 (3.6)	0.006
Frailty	137 (60%)	52 (63%)	0.496
ERAS	144 (63%)	46 (47%)	0.007

**Table 2 nutrients-14-00207-t002:** Results during hospital admission.

Variable	Controls (*n* = 227)	Cases (*n* = 98)	*p*-Value
Hospital stay	10.5 (7.2)	14.6 (14.1)	0.009
Adverse events	68 (32%)	42 (49%)	0.008
Medical complications	37 (16%)	30 (31%)	0.005
Surgical complications	62 (27%)	39 (40%)	0.036
Dehiscence	4 (2%)	4 (4%)	0.249
Surgical-wound infection	6 (3%)	10 (10%)	0.009
Paralytic ileus	54 (24%)	33 (34%)	0.076
Delirium	18 (8%)	8 (8%)	0.943
Transfusion	58 (26%)	42 (43%)	0.003
Number of concentrates	0.8 (2.0)	1.5 (2.4)	0.012
Readmissions	133 (59%)	67 (68%)	0.107
Reinterventions	19 (8%)	8 (7%)	0.457
ICU admission	6 (3%)	6 (6%)	0.195
In-patient mortality	9 (4%)	3 (3%)	0.939
365-day mortality	20 (9%)	12 (12%)	0.417

Legend: BMI: body mass index; ASA: American Society Anesthesiology scale; ERAS: Enhanced Recovery After Surgery program.

**Table 3 nutrients-14-00207-t003:** Univariate analysis variables associated with nutritional impairment estimated by CONUT score. Crude odds ratio and Charlson Comorbidity Index and ERAS odds ratios adjusted by age and sex. Univariate analysis variables associated with nutritional impairment estimated by CONUT score at hospital admission.

	Crude OR (95% CI)	*p*-Value	Adjusted OR (95% CI)	*p*-Value
Paralytic ileus	0.685 (0.414–1.131)	0.139	0.732 (0.435–1.233)	0.245
Adverse events	2.021 (1.212–3.372)	0.007	1.247 (1.127–3.333)	0.017
Surgery wound infection	4.186 (1.477–11.864)	0.007	5.780 (1.851–18.049)	0.003
Length of stay >12 days	2.004 (1.197–3.356)	0.008	1.184 (1.064–3.187)	0.029
In-hospital mortality	0.693 (0.203–2.888)	0.693	0.848 (0.217–3.314)	0.848
Transfusion	2.185 (1.327–3.600)	0.002	2.127 (1.221–3.704)	0.008
Medical complications	2.266 (1.300–3.948)	0.004	2.688 (1.475–4.897)	0.001
Surgical complications	1.759 (1.068–2.897)	0.026	2.317 (1.394–3.849)	0.001

**Table 4 nutrients-14-00207-t004:** Univariate analysis variables associated with nutritional impairment estimated by CONUT score at hospital discharge.

	Crude OR (95% CI)	*p*-Value	Adjusted OR (95% CI)	*p*-Value
Paralytic ileus	2.444 (1.463–4.084)	0.001	2.478 (1.457–4.214)	0.001
Adverse events	2.595 (1.600–4.209)	<0.001	2.673 (1.609–4.442)	<0.001
Surgery-wound infection	0.968 (0.355–2.645)	0.950	0.933 (0.324–2.690)	0.898
Urinary tract infection	9.172(1.149–73.247)	0.037	9.421 (1.095–81.04)	0.041
Length of stay >12 days	2.135 (1.286–3.544)	0.003	1.956 (1.155–3.313)	0.013
In-hospital mortality	12.468 (1.602–97.036)	0.016	14.668 (1.853–116.1)	0.011
Transfusion	3.573 (2.155–5.923)	<0.001	3.002 (1.751–5.146)	<0.001
Medical complications	2.633 (1.490–4.654)	0.001	2.730 (1.499–4.974)	0.001
Surgical complications	2.235 (1.377–3.627)	0.001	2.220 (1.340–3.680)	0.002

**Table 5 nutrients-14-00207-t005:** Bivariate analysis with nutritional status at hospital admission.

Variable	CONUT without Nutritional Disorder (*n* = 160)	CONUT with Nutritional Disorder (*n* = 167)	*p*-Value
Mortality	0 (0%)	13 (8%)	<0.001
Mortality 365 days	7 (4%)	26 (16%)	0.001
Adverse events	40 (27%)	72 (48%)	<0.001
Number of adverse events	1.1 (2.4)	2.1 (3.4)	0.003
Reintervention	9 (6%)	19 (11%)	0.076
Paralytic ileus	29 (18%)	58 (35%)	0.001
Suture dehiscence	4 (3%)	10 (6%)	0.172
ICU readmissons	2 (1%)	11 (7%)	0.020
Urinary tract infection	1 (1%)	9 (5%)	0.020
Surgery-wound infection	8 (5%)	8 (5%)	1.000
Transfusion	28 (18%)	74 (44%)	<0.001
Number of concentrates	0.4 (1.1)	1.6 (2.7)	<0.001
Medical complications	21 (13%)	47 (28%)	0.001
Surgical complications	35 (22%)	66 (40%)	0.001
ERAS	105 (66%)	86 (51%)	0.010
Frailty	93 (58%)	108 (65%)	0.256
Delirium	10 (6%)	16 (10%)	0.310

**Table 6 nutrients-14-00207-t006:** Survival mean (days) using the Kaplan-Meier analysis based on nutritional status estimated by CONUT. Analysis with the nutritional estimate using CONUT at hospital admission.

	Mean Estimation	95% Confidence Interval	
		Lower Limit	Upper Limit
No nutritional risk	341.437	327.837	355.036
Mild risk	359.870	349.887	369.852
Moderate risk	350.241	335.044	365.439
Severe risk	285.524	228.997	342.050

**Table 7 nutrients-14-00207-t007:** Analysis with the nutritional estimate using CONUT at hospital discharge.

	Mean Estimation	95% Confidence Interval	
		Lower Limit	Upper Limit
No nutritional risk	354.900	335.605	374.195
Mild risk	355.682	346.066	365.298
Moderate risk	338.787	322.305	355.270
Severe risk	297.806	249.505	346.108

## Data Availability

The study did not report any data.

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
