# Peer review of "Malnutrition and Increased Risk of Adverse Outcomes in Elderly Patients Undergoing Elective Colorectal Cancer Surgery: A Case-Control Study Nested in a Cohort"

_nutrients, 2022, doi:10.3390/nu14010207_

Round 1

Reviewer 1 Report

How was frailty assessed?

There are no references for some parameters evaluated such as the ASA score or ERAS.

The discussion is scarce, the results obtained should be commented more fully.

Author Response

Dear reviewer 1,

Please see the attachment with the responses. 

Best regards,

Reviewer 2 Report

This manuscript entitled Malnutrition and increased risk of adverse outcomes in patients undergoing elective colorectal cancer surgery: a case-control study nested in a cohort reported the poorer results of case patient with a CONUT score in the range of moderate or severe malnutrition (>4 points), comparing to the control patient, presenting a greater length stay, a higher incidence of adverse effects, etc. During the hospitalization an increased in malnutrition risk was observed and patients with severe malnutrition increased significantly mortality at 365 days. The authors conclude that malnutrition prevalence is high in this older adults (> 70 years) cohort and the prevalence increases during hospital admission which is linked to worse outcomes and suggest nutritional interventions will be critical in the perioperative period.

This manuscript presents high significance of content and good quality of presentation.

There are two comments:

  1. Suggest to add on elderly in the article title as Malnutrition and increased risk of adverse outcomes in elderly patients undergoing elective colorectal cancer surgery: a case-control study nested in a cohort.

2.In Table IV, please explain how to obtain the number of survival mean.

Author Response

This manuscript entitled Malnutrition and increased risk of adverse outcomes in patients undergoing elective colorectal cancer surgery: a case-control study nested in a cohort reported the poorer results of case patient with a CONUT score in the range of moderate or severe malnutrition (>4 points), comparing to the control patient, presenting a greater length stay, a higher incidence of adverse effects, etc. During the hospitalization an increased in malnutrition risk was observed and patients with severe malnutrition increased significantly mortality at 365 days. The authors conclude that malnutrition prevalence is high in this older adults (> 70 years) cohort and the prevalence increases during hospital admission which is linked to worse outcomes and suggest nutritional interventions will be critical in the perioperative period.

This manuscript presents high significance of content and good quality of presentation.

There are two comments:

Point 1. Suggest to add on elderly in the article title as Malnutrition and increased risk of adverse outcomes in elderly patients undergoing elective colorectal cancer surgery: a case-control study nested in a cohort.

Response 1: Elderly has been included in the title as reviewer # 2 suggested.

Point 2. In Table IV, please explain how to obtain the number of survival mean.

Response 2: Authors thank to reviewer #2 the comment. Days between brackets has been added to the title in order to clarify the survival mean obtaining.
